# A Novel lncRNA SAAL Suppresses IAV Replication by Promoting Innate Responses

**DOI:** 10.3390/microorganisms10122336

**Published:** 2022-11-25

**Authors:** Qingzheng Liu, Hongjun Yang, Lingcai Zhao, Nan Huang, Jihui Ping

**Affiliations:** 1MOE Joint International Research Laboratory of Animal Health and Food Safety, Engineering Laboratory of Animal Immunity of Jiangsu Province, College of Veterinary Medicine, Nanjing Agricultural University, Nanjing 210095, China; 2Key Laboratory of Livestock and Poultry Multi-Omics of MARA, Institute of Animal Science and Veterinary Medicine, Shandong Academy of Agricultural Sciences, Jinan 250100, China; 3Shandong Key Laboratory of Animal Disease Control and Breeding, Institute of Animal Science and Veterinary Medicine, Shandong Academy of Agricultural Sciences, Jinan 250100, China

**Keywords:** influenza A virus, long noncoding RNAs, SAAL, Serpina3i, IFN-β, ISGs

## Abstract

Influenza A virus (IAV) infection has traditionally been a serious problem in animal husbandry and human public health security. Recently, many studies identified that long noncoding RNAs play an important role in the antiviral immune response after the infection of the influenza virus. However, there are still lots of IAV-related lncRNAs that have not been well-characterized. Using RNA sequencing analysis, we identified a lncRNA, named Serpina3i Activation Associated lncRNA (SAAL), which can be significantly upregulated in mice after IAV infection. In this study, we found that overexpression of SAAL inhibited the replication of A/WSN/33(WSN). SAAL upregulated Serpina3i with or without WSN infection. Overexpression of Serpina3i reduced influenza virus infection. Meanwhile, knockdown of Serpina3i enhanced the replication of WSN. Furthermore, knockdown of Serpina3i abolished the SAAL-mediated decrease in WSN infection. Overexpression of SAAL or Serpina3i positively regulated the transcription of interferon β (IFN-β) and several critical ISGs after WSN infection. In conclusion, we found that the novel lncRNA SAAL is a critical anti-influenza regulator by upregulating the mRNA level of Serpina3i.

## 1. Introduction

Influenza A virus (IAV) belongs to the Orthomyxoviridae family. Based on 18 kinds of hemagglutinin (HA) proteins and 11 kinds of neuraminidase (NA) proteins, the influenza virus is divided into different subtypes. The IAV genome consists of eight single-stranded, negative-strand RNAs, which encode 18 proteins [1,2,3,4,5]. The influenza A virus is under a wide host spectrum and can cause acute respiratory diseases in humans, poultry, pigs, cattle and many other animals. Therefore, the influenza virus has caused pandemics in various regions, causing casualties and major economic losses. It is part of the important zoonotic diseases in the world [6,7,8].

Noncoding RNAs (ncRNAs) mainly include miRNAs, tRNAs, cirRNAs and lncRNAs. They are new transcripts with a genome-coding domain, but most of them are not translated into proteins [9,10,11]. At present, it has been recognized that these ncRNAs are key regulatory factors in the interaction between influenza virus and host, which were associated with regulating the production of transcription factors, interferons, cytokines and ISGs [12]. lncRNAs are kinds of nonprotein-coded transcripts with a length of more than 200 nucleotides, and played an important role in biological processes, such as genomic imprinting, development, stem cell versatility and so on. lncRNAs also affect the development of diseases, such as cancer metastasis, atherosclerosis and inflammatory response, and they are closely related to the immune system [13,14,15,16,17]. Although some lncRNAs have been found to play important roles in the antiviral immune process, most functions of lncRNAs are still not clear.

Previous studies have revealed that some lncRNAs act as antiviral regulators in innate immune response [18]. Furthermore, lncRNAs have a variety of biological activities and a wide range of regulatory mechanisms, indicating that lncRNAs are crucial in the natural cellular immunity caused by viral infection. For example, lncRNA NeSTRNA regulates the transcription of the gene encoding IFN-γ [19]. After influenza virus infection in A549 cells, the expression level of lncRNA NRAV could significantly upregulate expression. NRAV regulates the dehistone modification of promoters by interacting with ZONAB, so as to inhibit the transcription of IFITM3 and MxA, and further facilitates the replication of IAV [20]. Interaction between lncRNA PAAN and IAV PA protein promotes RNA polymerase assembly and provides efficient synthesis of IAV RNA, thereby promoting viral replication [21]. The expression of lncRNA PSMB8-AS1 significantly increases after IAV infection and IFN-β stimulation. Inhibition of PSMB8-AS1 expression reduces the expression level of viral mRNA and protein, thus attenuating IAV particles’ release [22]. The IFN-induced lnc-Lsm3b is inactivated RIG-I in innate immune response [23], and lncRNA ACOD1 facilitates the replication of the influenza virus by regulating cell metabolism [24]. lncRNA ISG20 is upregulated in A549 cells and 293 T cells after infection with IAV. Overexpression of lncRNA ISG20 inhibits IAV replication [25]. In summary, lncRNAs plays prominent roles in influenza virus infection.

In this study, we identified a novel lncRNA whose mRNA level was significantly increased after IAV infection both in vivo and in vitro. This lncRNA elevated the mRNA level of Serpina3i, so it was named Serpina3i Activation Associated lncRNA (SAAL). We found that SAAL suppressed IAV replication by positively regulating the transcription of IFN-β and several critical ISGs. Our observations clarified that lncRNA SAAL is a positive regulator in host antiviral defense system.

## 2. Materials and Methods

### 2.1. Cell Lines and Virus

Madin–Darby canine kidney (MDCK) cells, human alveolar epithelial A549 cells and human embryo kidney 293T cells were maintained in Dulbecco’s modified Eagle’s medium (Gibco, Invitrogen, Carlsbad, CA, USA) supplemented with 10% fetal bovine serum (FBS) (Gibco, Invitrogen, Carlsbad, CA, USA). Mouse lung epithelium Mle-12 cells (ATCC CRL-2100) were maintained in Dulbecco’s modified Eagle’s medium/F-12 medium added with 2% FBS and antibiotics. All cells were maintained at 37 °C in 5% CO_2_. The low pathogenic H7N9 subtype AIV (A/Anhui/1/2013) used in this study was rescued by the reverse genetics system [26]. A/WSN/1933(H1N1), H9N2 subtype AIV (A/chicken/Guangxi/55/2005) and Sendai virus (SeV) were propagated in specific pathogen-free (SPF) embryonated chicken eggs. 

### 2.2. Virus Infection 

Female 4- to 6-week-old BALB/c mice were inoculated intranasally with 5 × 10^5^ EID50 A/Anhui/1/2013 influenza A virus. Then, 3 days post infection, mouse lungs were collected for further analysis. All animal work was carried out in an enhanced BSL2 + laboratory. The researchers working in the BSL2 + lab wore Tyvek and powered purified air respirators. Mle-12 cells were cultured approximately 18 h in 12-well plates. The cells were infected with influenza virus at 0.5 multiplicity of infection (MOI). Infected cells were cultured with serum-free medium contained at 37 °C and 5% CO_2_ incubator for 1 h. The cells were then washed with phosphate-buffered saline (PBS) twice, and serum-free medium containing TPCK-Trypsin (0.1 µg/mL) was added. Then, 24 h post infection, cell culture supernatants were collected to confirm virus titer by hemagglutination assay and plaque assay. 

### 2.3. qRT-PCR Analysis

Total RNA was isolated from cells using Trizol (Invitrogen, Carlsbad, CA, USA). The cDNA was generated using the HiScript II 1st Strand cDNA Synthesis Kit (Vazyme, R211-02). After cDNA synthesis, qPCR was conducted by AceQ qPCR SYBR Green Master Mix (Vazyme, Q111-02). Primers used in qRT-PCR are listed in Appendix A. 

### 2.4. Hemagglutination Assay

The cell supernatants were diluted with PBS and mixed with an equal volume of 1% chicken erythrocytes. The virus titer was calculated from the highest dilution factor that produces a positive reading [27]. 

### 2.5. Plaque Assay

MDCK cells were cultured in 6-well plates and incubated with serial dilutions of virus for 1 h. After incubation, the cells were supplemented with DMEM containing 1% agarose and 1µg/mL of TPCK-trypsin. After agarose solidified, the cells were incubated upside-down at 37 °C for 48–72 h. Then, the virus plaques were counted and PFUs were determined. 

### 2.6. Bioinformatics Analysis of Noncoding Potential

Noncoding potential of lncRNA SAAL was analyzed by coding-potential calculator [28].

### 2.7. Statistical Analysis

The Student’s t-test was used to analyze the statistical comparison between the two groups. ns = not significant, * *p* < 0.05, ** *p* < 0.01 *** *p* < 0.001, **** *p* < 0.0001. Error bars represent standard error (± SD). Statistical analysis was performed using GraphPad Prism version 8.0.

## 3. Results

### 3.1. IAV Infection Induces lncRNA SAAL Expression in Mice

To study the function of host lncRNAs in response to IAV infection, we used lncRNA microarrays to analyze altered expression of lncRNAs in BALB/c mice infected with or without the A/Anhui/1/2013(H7N9) virus. The RNA sequencing data was presented in previous studies [29]. The schematic description of the RNA-seq experimental design are shown in Figure 1A. Using a *p*-value of <0.05, 7045 upregulated lncRNAs and 981 downregulated lncRNAs were changed 2-fold or more in the lung tissues of infected mice in comparison with uninfected mice (Figure 1B). Based on these data, we selected 20 lncRNAs whose expression was significantly changed (Figure 1C). Then, they were selected for further confirmation by qRT-PCR. The results were consistent with the trend of sequencing data (Figure 1D).

To verify whether these 20 lncRNAs can affect the replication of IAVs, we constructed these lncRNA-pcDNA3.1(+) expressing vectors. Then, we transfected these plasmids into Mle-12 cells for 24 h. After that, the cells were infected with the influenza virus A/WSN/33 for 24 h. The cell supernatants were collected for hemagglutination assay (HA). The HA results showed that during these lncRNAs, only lncRNA SAAL (Gene ID: TCONS_00373113) could suppress the replication of A/WSN/33 virus (Figure 2A). Plaque assay data are shown in Figure 2B. The virus titers in the supernatants obtained from lncRNA SAAL-overexpressed cells were significantly lower than those from the empty vector (EV) control cells. The results of the Western blot analysis were consistent with the above (Figure 2C). These data suggest that overexpression of lncRNA SAAL suppressed the replication of IAV. 

The lncRNA SAAL is located on mouse chromosome 16, and bioinformatics analysis of noncoding potential was performed by Coding Potential Assessment Tool (Table 1). The results suggested that it is a noncoding transcript [28].

In this study, we observed that lncRNA SAAL was upregulated after WSN infection in Mle-12 cells (Figure 3A). Furthermore, SAAL was significantly increased by poly(I:C) (Figure 3B).

Next, we examined whether type I IFN can induce the expression of lncRNAs as identified above. We treated Mle-12 cells with 20 ng/mL and 100 ng/mL IFN-β for 6 h; then, we detected the mRNA level of lncRNAs by real-time PCR [30]. The data indicated that the mRNA level of SAAL was significantly increased in IFN-β-treated cells (Figure 3C). These results demonstrated that SAAL is an interferon-stimulated gene.

### 3.2. Serpina3i Is Coexpressed with SAAL

To further understand the mechanism of how SAAL suppress IAV replication, we performed a coexpression analysis of several lncRNAs to predict their protein partners (Figure 4A). Interestingly, these results showed that SAAL and the other three candidate lncRNAs (ID: NONMMUT109312.1, NONMMUT058145.2 and NONMMUT145028.1) possess the same predicted mRNA, Serpina3i (ID: NM_001199940.1). According to the coexpression analysis, we selected eight antiviral immune responses-related mRNAs as candidates. Then, 12 h after SAAL was overexpressed in Mle-12 cells, we detected their mRNA level by real-time PCR (Figure 4B). These results indicated that SAAL significantly promoted the transcription of Serpina3i, and there was no significant influence compared to the other seven candidates. Thus, we concluded that Serpina3i is coexpressed with SAAL.

Furthermore, we wished to further explore the function of Serpina3i during the antiviral process. Next, we constructed a Serpina3i-pcDNA3.1(+) expressing vector, then tested its effect on IAV replication depending on the methods and conditions mentioned above. As well, we detected the viral titer in Serpina3i-expression cells after infection of A/WSN/1933 by plaque assay (Figure 5A). These results indicated that overexpression of Serpina3i inhibits the replication of A/WSN/1933. The results of the Western blot analysis were consistent with the above (Figure 5E). Next, we knocked down Serpina3i in Mle-12 cells by RNA interference and analyzed the IAV infection as described above. These cells were treated with small interfering RNA (siRNA) and reduced the mRNA level of Serpina3i (Figure 5B). Plaque assay indicated that virus titer was significantly higher in cells with decreased amounts of Serpina3i than in control cells (Figure 5C). These results demonstrated that Serpina3i elicited an inhibitory effect on IAV replication. To further determine the relationship between SAAL and Serpina3i, Mle-12 cells were transfected with si-NC or si-Serpina3i for 24 h and then transfected with pcDNA3.1(+) or SAAL- pcDNA3.1(+) for 12 h [31]. Cells were then infected with A/WSN/1933 at MOI = 0.5, and virus titer in cell culture was measured at 24 h post infection. Consistently, SAAL lost its antiviral ability post si-Serpina3i treatment in Mle-12 cells (Figure 5D). Taken together, SAAL inhibited the replication of IAV by eliciting the mRNA level of Serpina3i. Then, we observed that Serpina3i was upregulated after WSN infection in Mle-12 cells (Figure 5F). As well, Serpina3i was significantly increased by poly(I:C) (Figure 5G). Next, we treated Mle-12 cells with 20 ng/mL and 100 ng/mL IFN-β for 6 h; then, we detected the mRNA level of Serpina3i. The data indicated that Serpina3i was significantly increased in IFN-β-treated cells (Figure 5H).

### 3.3. SAAL- and Serpina3i-Induced mRNA Level of IFN-β and ISGs

Next, we wonder if SAAL and Serpina3i affects the replication of IAV by interfering with the antiviral immune response. We first examined the effect of SAAL and Serpina3i on the mRNA level of IFN-β in WSN infected cells. The data from qRT-PCR showed that the overexpression of SAAL and Serpina3i upregulated the mRNA level of IFN-β in 6 h after WSN infection (Figure 6A). However, at 12 h after viral infection, the mRNA level of IFN-β in the SAAL overexpression group was not significantly upregulated compared with that in the control group (Figure 6B). These results indicated that SAAL and Serpina3i could promote the transcription process of IFN-β in early stages of WSN infection. Under the same conditions, we measured the mRNA levels of ISGs, such as ISG15, IFIT1 and IFIT2. The data showed that the mRNA level of ISGs in SAAL overexpression and Serpina3i overexpression cells was significantly higher than that in control cells (Figure 6C,D).

## 4. Discussion

Influenza A virus has historically been a serious threat to the animal breeding industry and human health [32]. Recent years, a lot of lncRNAs have been identified; however, their antiviral mechanisms remain unclear. Type I interferons play important roles in antiviral innate immune response [33]. Meanwhile, many lncRNAs are identified as ISGs to regulate virus replication. For example, lnc-ISG20 can be stimulated by IFN-β and suppress IAV replication by competitively binding to miR-326 to enhance ISG20 expression [25]. lncRNA ISR, an interferon-stimulated lncRNA, is regulated by RIG-I-dependent signaling that influences IFN-β production to inhibit IAV replication [34].

In this study, we showed that 8195 lncRNAs were differentially expressed after IAV infection. Among these, 8026 lncRNAs were upregulated and 169 lncRNAs were downregulated. According to the criteria of *p*-value < 0.05 and multiple of difference > 20, a total of 20 candidate genes was selected based on the fold changes. The expression trend of the candidate genes was verified by qRT-PCR. These results were consistent with the transcriptome sequencing analysis; then, we constructed these 20 candidate genes into a pcDNA3.1(+) vector; thus, we transfected them into mouse lung epithelial cells (Mle-12). After overexpression for 24 h, A/WSN/1933 (H1N1) was infected with a dose of MOI = 0.5. The cell supernatant was collected 24 h after virus infection, and the preliminary function was verified by the hemagglutination test. The verification result found that one of the candidate genes (ID: TCONS_00373113) can significantly inhibit the replication of the WSN virus. The virus titer of the collected samples was titrated by the plaque assay, and the trend was consistent with that of the hemagglutination assay results. Since lncRNA SAAL was extremely low-expressed in healthy mouse lung epithelial cells, it could not be verified for knockdown efficiency after synthesizing small interfering RNA. Therefore, we concluded that this was an antiviral gene, and we named it SAAL, based on its mechanism of action. Then, we discovered that the candidate gene did not have the coding ability through the prediction results of the online website Coding Potential Assessment Tool. In order to further explore the antiviral mechanism of SAAL, we carried out lncRNA–mRNA coexpression prediction analysis, and then made an interesting discovery: the prediction results of four candidate lncRNAs contained identical coexpression mRNA (Serpina3i). Then, we screened out eight candidate target genes according to the coexpression correlation and the role of mRNA. Results of qRT-PCR showed that the transcription level of Serpina3i was significantly upregulated after overexpression of SAAL. Therefore, it can be concluded that there is a coexpression relationship between Serpina3i and SAAL. In order to verify the function of Serpina3i, we constructed a Serpina3i-pcDNA3.1 overexpression plasmid and synthesized small interfering RNA (si-Serpina3i). Verified as described above, we found that overexpression of Serpina3i inhibited the replication WSN, and knockdown of Serpina3i promoted the replication of the WSN virus. These results showed that Serpina3i can inhibit the replication of the influenza virus. Our study first revealed the antiviral function of Serpina3i. In order to explore the relationship between Serpina3i and SAAL, we supplemented SAAL after knockdown of Serpina3i. The results showed that on the premise of knockdown of Serpina3i, supplementing SAAL did not inhibit the replication of WSN. Therefore, we conclude that SAAL inhibited the replication of the influenza virus by upregulating the transcription level of Serpina3i, and Serpina3i is the major regulatory gene of SAAL. Later, in order to further study the antiviral mechanism of SAAL and Serpina3i, we infected WSN 24 h after overexpression of SAAL or Serpina3i in Mle-12 cells and collected cell lysates 6 h and 12 h after infection. The results showed that the transcription level of IFN-β was significantly upregulated after overexpression of SAAL or Serpina3i. Subsequently, we detected the mRNA levels of several classical ISGs and found that they were also upregulated. According to this study, we can determine that SAAL upregulates the level of IFN transcription in some way. However, the specific regulation mode is still unclear. It may participate in the regulation of PRRs, such as RIG-I, MDAS, TLRs, etc., or participate in the regulation of transcription factors. Therefore, the mechanism of SAAL regulation of IFN remains to be further studied.

In summary, our study discovered a novel lncRNA, SAAL, which can promote the transcription level of Serpina3i so as to upregulate the mRNA level of IFN-β and ISGs, thereby inhibiting the replication of the influenza virus. Our study revealed the antiviral function of lncRNA SAAL and Serpina3i for the first time. This study provides more evidence for host lncRNAs to participate in inhibiting influenza virus replication, which will contribute to the design of drugs for the treatment of the influenza virus.

## Figures and Tables

**Figure 1 microorganisms-10-02336-f001:**
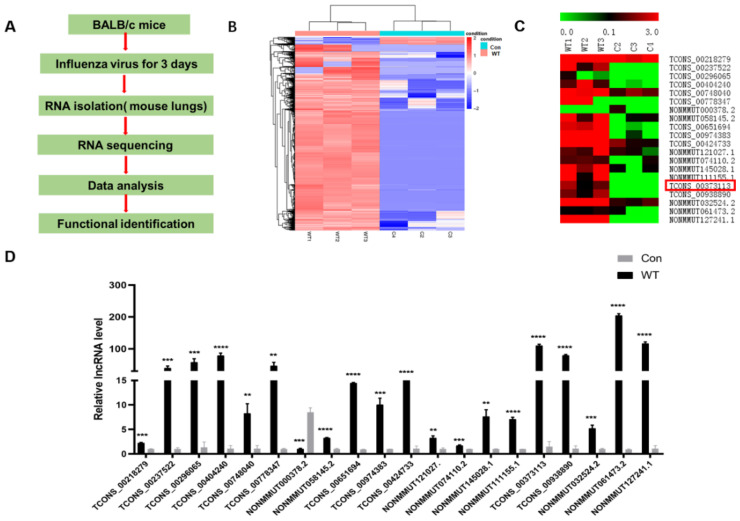
Different regulated lncRNAs after influenza virus infection. (**A**) Schematic description of the RNA-seq experimental design. (**B**) Analysis of RNA sequencing of mouse with IAV infection and after recovery revealed 7045 upregulated and 981 downregulated lncRNAs (*p* < 0.05, fold change > 2). (**C**) 20 lncRNAs are selected and shown in heat maps. (**D**) The relative levels of selected lncRNAs in the infected mouse were examined by RT-qPCR (*n* = 3; means and SD; ** *p* < 0.01 *** *p* < 0.001, **** *p* < 0.0001).

**Figure 2 microorganisms-10-02336-f002:**
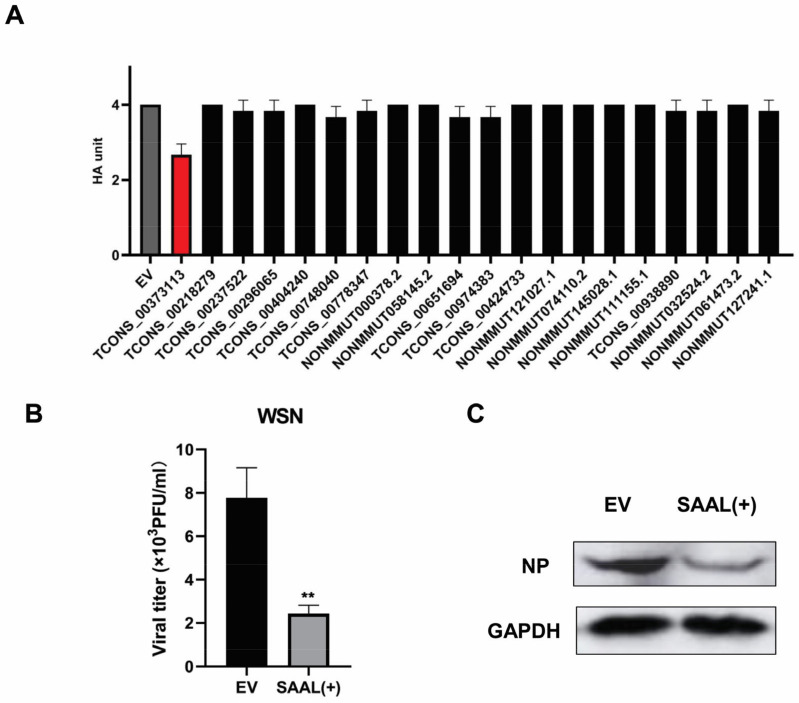
SAAL suppresses IAV regulation in Mle-12 cells. (**A**) HA results of overexpressed lncRNAs together with the control after WSN infection (MOI = 0.5). The viral titer in the supernatant was measured 24 h post infection (hpi). Shown are means for three independent experiments ± SD (*n* = 3; ** *p* < 0.01). (**B**) Plaque assay evaluated the IAV replication after the overexpression of SAAL in Mle-12. Viral titers in supernatants were measured at 24 hpi. The means for three independent experiments ± SD are shown (*n* = 3; ** *p* < 0.01). (**C**) Western blotting evaluated the IAV replication after the overexpression of SAAL in Mle-12 cells.

**Figure 3 microorganisms-10-02336-f003:**
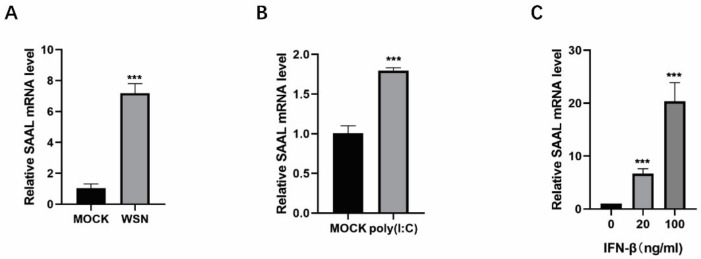
SAAL is an interferon-stimulated gene and can be induced by several viruses. (**A**) Mle-12 cells were infected with WSN, and RT-qPCR was performed to determine the mRNA level of SAAL. The means for three independent experiments ± SD are shown (*n* = 3; means and SD; *** *p* < 0.001). (**B**) Mle-12 cells were stimulated by poly(I:C). The means for three independent experiments ± SD are shown (*n* = 3; means and SD; *** *p* < 0.001). (**C**) Mle-12 cells were stimulated with increasing amounts of IFN-β for 16 h. RT-qPCR was performed to determine the levels of SAAL. The means for three independent experiments ± SD are shown (*n* = 3; means and SD; *** *p* < 0.001).

**Figure 4 microorganisms-10-02336-f004:**
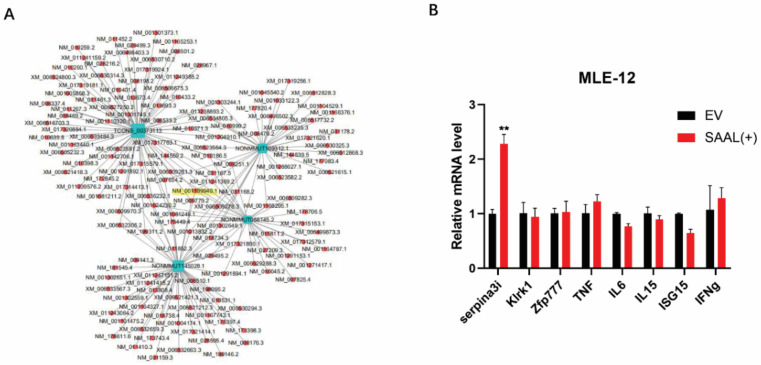
Bioinformatic analysis and RT-qPCR identified that Serpina3i is coexpressed with SAAL. (**A**) lncRNA-mRNA coexpression map. lncRNA was represented by a blue square, and mRNA was represented by a red circle. ID of Serpina3i was highlighted by yellow color. (**B**) RT-qPCR was performed to determine the levels of candidate mRNAs. The means for three independent experiments ± SD are shown (*n* = 3; means and SD; ** *p* < 0.01).

**Figure 5 microorganisms-10-02336-f005:**
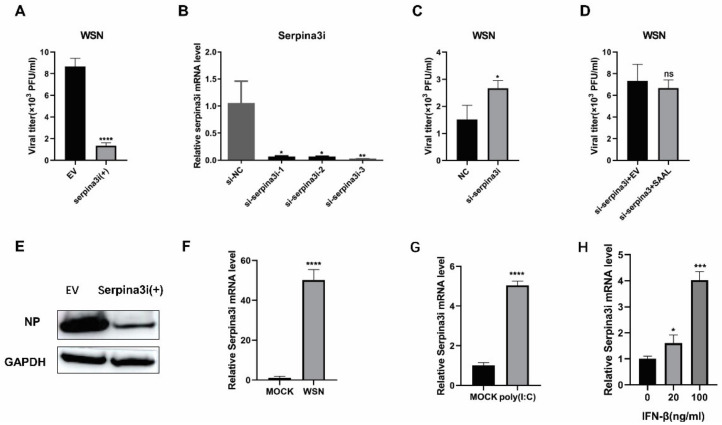
Serpina3i suppresses IAV regulation in Mle-12 cells. (**A**) Plaque assay evaluated the WSN replication after the overexpression of Serpina3i in Mle-12. The means for three independent experiments ± SD are shown (*n* = 3; **** *p* < 0.0001). (**B**) The mRNA level of Serpina3i in Mle-12 cells after being transfected with siRNAs. The means for three independent experiments ± SD are shown (*n* = 3; * *p* < 0.05; ** *p* < 0.01). (**C**) Plaque assay evaluated the WSN replication after the knockdown of Serpina3i in Mle-12. The means for three independent experiments ± SD are shown (*n* = 3; * *p* < 0.05). (**D**) Mle-12 cells were transfected with si-Serpina3i for 24 h and then transfected with pcDNA3.1 or pcDNA-SAAL for 12 h. Viral titers in supernatants were measured at 24 hpi. (**E**) Western blotting evaluated the IAV replication after the overexpression of Serpina3i in Mle-12. (**F**,**G**) Mle-12 cells were stimulated with WSN and poly(I:C). Then. RT-qPCR was performed to determine the mRNA level of Serpina3i. (**H**) Mle-12 cells were stimulated with increasing amounts of IFN-β for 16 h. RT-qPCR was performed to determine the levels of Serpina3i. The means for three independent experiments ± SD are shown (*n* = 3; * *p* < 0.05, *** *p* < 0.001).

**Figure 6 microorganisms-10-02336-f006:**
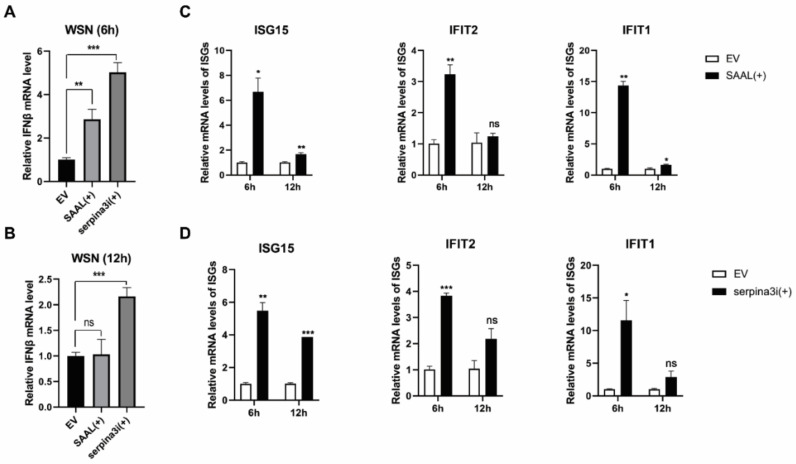
Overexpression of SAAL and Serpina3i in Mle-12 cells changes the expression of IFN-β and several critical ISGs. (**A**) SAAL and Serpina3i were transiently overexpressed in mle-12 cells, and the mRNA level of IFN-β was detected at 6 h post WSN infection. (**B**) SAAL and Serpina3i were transiently overexpressed in mle-12 cells, and the mRNA level of IFN-β was detected at 12 h post WSN infection. (**C**) The mRNA levels of several critical ISGs were detected after the overexpression of SAAL. (**D**) The mRNA levels of several critical ISGs were detected after the overexpression of Serpina3i. The means for three independent experiments ± SD are shown (*n* = 3; means and SD; * *p* < 0.05, ** *p* < 0.01 *** *p* < 0.001).

**Table 1 microorganisms-10-02336-t001:** Coding ability prediction of lncRNA SAAL.

lncRNA ID	RNA Size	ORF Size	Ficket Score	Hexamer Score	Coding Probability	Coding Label
TCONS_00373113	544	186	0.904	−0.239	0.041	no

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
