# Peer review of "A Novel lncRNA SAAL Suppresses IAV Replication by Promoting Innate Responses"

_microorganisms, 2022, doi:10.3390/microorganisms10122336_

Round 1
Reviewer 1 Report
1. Line 31 "Influenza A virus is under a wide host spectrum" should be "Influenza A virus was under a wide host spectrum"
2.Line 108 "2.6. HI assay and antigenic cartography" should be deleted
3.Does this lncRNA SAAL locate in nucleus or cytoplasm?
4. You didn't verify that if knockdown of SAAL can influence the replication of WSN, please tell me why not.
Reviewer 2 Report
This manuscript from Liu et al details a novel role of the lncRNA labeled SAAL by the investigators and its target mRNA (Serpina3i) inhibit IAV replication in-vitro, likely by stimulating early interferon production. SAAL was identified as a lncRNA candidate of interest by RNA sequencing following IAV infection in mice, and its role in antiviral control was investigated, and its role in anti-viral control was identified through its expression in IAV infected MLE-12 cells. Notably, the expression of SAAL and Serpina3i along with 3 other lncRNAs targeting Serpina3i were overexpressed during IAV infection, and the anti-viral properties of Serpina3i were also demonstrated in-vitro.
This work establishes SAAL as another host lncRNA regulating IAV replication. I have a few comments to be addressed prior to acceptance for publication:
1) Unlike SAAL, most of the host lncRNAs mentioned in the introduction (page 2) seemed to be anti-inflammatory. The authors should briefly comment on whether these lncRNAs are expressed and act later during viral infection, as this may differentiate them from SAAL.
2) On page 3 lines 120-121, a cited manuscript was incorrectly labeled as a clinical trial when these studies were performed in mice. This should be corrected.
3) The y axis in Figure 2A should be changed since these results were from a hemagglutination assay instead of a plaque assay, and therefore not a direct measure of viral titer.
4) The other lncRNA candidates that target Serpina3i on page 6 line 181 were not included in figure 1- it would be nice to have them included to see if they have similar behavior to SAAL.
5) The authors should determine how SAAL affects Serpina3i expression, especially considering that most of its antiviral effects are seen early in infection while effects on mRNA stability might not be seen until later timepoints.
6) The discussion section should expand more on their results and put them into a wider context instead of reiterating them. Some suggestions for discussion topics include possible expression mechanisms of SAAL to allow for its effects to be seen early during infection, therapeutic potential of SAAL and other host lncRNAs and whether IAV may counteract host lncRNAs.
Round 2
Reviewer 2 Report
All of the comments were addressed- the only comment I have before publication is to change "clinical trial" on page 3 line 121 to "the RNA sequencing data."